



# Thundercloud structures detected and analyzed based on coherent Doppler wind lidar

Kenan Wu[1], Tianwen Wei[1,2], Jinlong Yuan[1,2], Haiyun Xia[1,2,3,4,5,6], Xin Huang[1], Gaopeng Lu[1], Yunpeng Zhang[1], Feifan Liu[1], Baoyou Zhu[1], and Weidong Ding[7]

[1]School of Earth and Space Science, University of Science and Technology of China, Hefei 230026, China
[2]School of Atmospheric Physics, Nanjing University of Information Science and Technology, Nanjing 210044, China
[3]Hefei National Laboratory for Physical Sciences at the Microscale, Hefei 230026, China
[4]Institute of Software, Chinese Academy of Sciences, Beijing 100190, China
[5]Collaborative Innovation Center on Forecast and Evaluation of Meteorological Disasters, Nanjing University of Information Science and Technology, Nanjing 210044, China
[6]Aerosol-Cloud-Precipitation Key Laboratory, NUIST, CMA, Beijing 100081, China
[7]Anhui Meteorological Observatory, Hefei 230031, China

*Correspondence to*: Haiyun Xia (hsia@ustc.edu.cn)

**Abstract.** The studies of intracloud (IC) discharges might shed light on the microphysical structure of thunderclouds. As both the magnitude and the sign of charge separation due to graupel collides with ice crystals within the strong updrafts are influenced by the surrounding environment. Here, a compact all-fiber coherent Doppler wind lidar (CDWL) working at the 1.5 μm wavelength is applied for probing the dynamics and microphysics structure of thunderstorms. Thanks to the precise spectrum measurement, multi-component spectra signals of thunderstorms can be analyzed by the CDWL. The spectrum width, skewness, and Doppler velocity of CDWL is used to separate and identify the particle composition and polarity. In experiment, the thundercloud properties are detected by the CDWL, 10.6 cm Doppler weather radar (DWR), and Advanced Geosynchronous Radiation Imager (AGRI) onboard Fengyun-4 satellites. In particular, the spectrum width and skewness of the thundercloud below the 0 °C isotherm are increased, and when a cloud-ground lightning occurs, there has additional graupel with a velocity greater than 5 m/s. It indicates that this region is a melting layer, and lightning activity changes the motion characteristics of graupel, affecting the charge structure of the whole thundercloud. In general, our findings provide details on the velocity, phase, and composition of particles in the outside updraft region of the thunderstorm. The identification and analysis of graupel is particularly important. It is proved that the precise spectrum of CDWL is a promising indicator to study the charge structure of thunderstorms.

## 1 Introduction

The co-development of precipitation particles and electric charge during the initial electrification of thunderstorms has been investigated for more than a century. Graupel particles are critically important in this process (Goodman et al., 1988). The electrification of thunderstorms first requires the existence of a mixed-phase updraft inside, resulting in ice-graupel collisions. Graupel gains charge in the presence of liquid water, which is called the non-inductive charging (NIC) mechanism (Jayaratne



et al., 1983; Saunders, 2008; Takahashi et al., 2017). Larger cloud particles (graupel or hail) fall under gravity while smaller particles (ice crystals or small water drops) are transported in the updraft. These particles could affect thunderstorm structure, microphysics, and lightning activity under different environmental conditions (Carey and Rutledge, 2000; Carey and Buffalo, 2007; Fuchs et al., 2018; Bruning et al., 2007; Dye et al., 1989).

The NIC mechanism is thought to be primarily responsible for the thunderstorm discharges. The theory is based on experiments conducted in Japan by Takahashi (1978) and in the UK by Jayaratne et al. (1983). Observations revealed that the magnitude and polarity of the charging process depend on the water content of the cloud and the ambient temperature. During collisions graupel charge positively at higher temperatures, at both low and high water content, and they charge negatively at low temperatures and at intermediate water content. Consider a typical cloud with a liquid water content of approximately 1 $g/m^3$, the positive charge center is located above the negative one, and the negative one is very shallow, approximately 1 km in thickness, and located in a region of -15 to -10 ℃ isotherm. Below the negative charge center is a small positive charge pocket. As the graupel particles fall from greater heights through the clouds, they collide with ice crystals that are being carried upward in updrafts. If the temperature is below approximately -15 to -10 ℃, the graupel particles charge negatively and the ice crystals positively. The light positively charged ice crystals travel upward along the updraft, leaving the positive charge at a higher location in comparison with the negatively charged falling graupel particles. As the graupel particles fall further, the temperature increases and the graupel particles start to charge positively. Thus, there is a region below the height of the isotherms -15 to -10 ℃ where graupel particles are positively charged. This is the basis of the positive charge pocket located below the negative charge center. This creates the observed tripolar structure of the cloud (Bruning et al., 2014; Williams, 1989, 2001; Bruning et al., 2010). It also explains why the main negative charge center is located in the region of the -15 and -10 ℃ isotherm.

The dependence of charge structures on cloud water content and ambient temperatures has sparked interest in detecting cloud environment changes in the mixed phase regions during whole storm processes (Bruning et al., 2010; Fuchs et al., 2018). In both observations and simulations, the supercell produced a complex evolution of charge structure with six to seven different layers frequently existing with a relatively larger horizontal extent at different cloud water content and ambient temperatures (Calhoun et al., 2014). Some severe storms have a consistent dominant upper-level inverted dipole charge structure near the updraft, which has a low cloud liquid water content (Coquillat et al., 2022; Lang et al., 2004). Bruning et al. (2014) analyzed inverted polarity thunderstorms and showed that the variability in NIC generation mechanism could continuously alter the electrification and charge structure in strong updraft.

Lightning activity is closely related to thunderstorm structural parameters (Cheng et al., 2022; Sun et al., 2021). The spatial and temporal details of the area participating in lightning discharges in the cloud are still being revealed. For example, most of the stratiform lightning are closely related to the melting layer (Wang et al., 2019b). During mei-yu period, strong updrafts transports supercooled liquid water into to the mixed phase region between 0 and −10°C levels and producing cloud-ground (CG) lightning (Yang et al., 2022). The distribution of lightning relative to altitude and radar reflectivity varies with lightning and storm type (Mecikalski and Carey, 2018).



However, the knowledge about the electrification process and life cycle in thunderstorms is still limited. Using the very high frequency (VHF) source emissions detected by the lightning mapping array, the polarity, height, and thickness of vertical charge distribution are estimated (Medina et al., 2021; Fuchs et al., 2015; Lang and Rutledge, 2011; Zhang et al., 2022; Fuchs et al., 2016; Figueras I Ventura et al., 2019b; Erdmann et al., 2020). Satellites are widely used to study the effects of ice processes on the microphysical, dynamic, and thermodynamic development in mixed-phase clouds and their interactions with aerosols (Chen et al., 2020; Zhang et al., 2022; Sassen and Wang, 2008; Khanal and Wang, 2018). Radars play an essential role in hydrometeor identification and calculation of microphysical properties in all storms (Fan et al., 2018; Stough et al., 2021; Li et al., 2020; Wang et al., 2022a). At present, high-resolution severe storm observations obtained by vehicle-mounted mobile X-band dual-polarization radar are widely used in severe storm structure and dynamics studies (Zhao et al., 2020; Stolzenburg et al., 2015; Qie et al., 2021; Figueras I Ventura et al., 2019a). Observed thunderstorm charge structures are often varied and complex in time and space. More detection methods need to be developed to obtain details of the cloud phase transformation, ice crystal evolution, and charge transfer with high spatial and temporal resolution.

In this study, a coherent Doppler wind lidar (CDWL) system is used to detect and analyse the development of thunderstorm over Hefei, China. The CDWL, as an active optical remote sensing instrument, measures the radial velocity accurately. It has been applied for research on the atmospheric boundary layer height (Wang et al., 2019a; Wang et al., 2021), gravity waves (Jia et al., 2019), turbulence (Wang et al., 2022b; Smalikho and Banakh, 2017; Banakh et al., 2021), windshear (Yuan et al., 2020; Yuan et al., 2022a), precipitation (Wei et al., 2019; Wei et al., 2021), air pollution (Yuan et al., 2022b), and bioaerosol transport (Tang et al., 2022). Recently, the CDWL has been extended to simultaneously detect the aerosol and melting snow signals of a melting layer (Wei et al., 2022) and the cloud water/ice signals during cloud seeding (Yuan et al., 2021). It provides high spatial and temporal resolution results of atmospheric composition by the deep analysis of the power spectrum. The CDWL can detect cloud environments, reflect cloud phases and has the potential to detect the intracloud (IC) charge structure in the thunderstorm through accurate spectral measurements, thus it can be developed as a new instrument for thunderstorm detection.

This paper is organized as follows: the instruments and datasets are described in Sect. 2. Section 3 introduces the CDWL products derived from power spectrum. Section 4 presents the observation results from CDWL, FY-4 and DWR. Through the deep analysis of the power spectrum, distribution of different particles in the thundercloud and the velocity change of graupel during the lightning occurs detected by CDWL are discussed. Finally, a conclusion is drawn in Section 5.

## 2 Instruments and datasets

### 2.1 Ground-based instruments

The datasets from two ground-based remote sensing instruments are utilized in the study. The CDWL is installed on the roof of a 63.8-meter-high building (31.841 °N, 117.270 °E) on the campus of the University of Science and Technology of China (USTC). The Doppler weather radar (DWR) is located 3.5 km away in the northwest. The two instruments are shown on the



right of Figure 1. The key parameters are listed in Table 1. Besides, a ground-based optical disdrometer (second-generation

particle size and velocity, Parsivel-2 (Tokay et al., 2014)), a Davis weather station (wireless vantage pro2 plus) and a Micro-Electro-Mechanical System (MEMS)-atmospheric ground $E$-field sensor are also deployed for comparative measurements about 50 m away from the CDWL.

The compact all-fiber CDWL operates with an eye-safe wavelength of 1.5 μm. The pulse duration and pulse energy of the laser are 600 ns and 300 μJ, respectively. The radial spatial resolutions are set at 30/60/150 m in the range of 0–2.22/2.22–

5.22/5.22–12.72 km. The range-varying resolution is designed to improve the detection probability at high heights where the aerosol concentration is low. During the experiments, the CDWL work in velocity-azimuth display (VAD) scanning mode for wind profile detection. It is an extension of a staring mode, where the lidar beam rotates around a vertical axis, thus forming a cone with the base at the measurement distance of interest and the apex at the lidar source (Sathe and Mann, 2013; Banakh et al., 2017). The azimuth scanning range is set as 0-300 ° and the elevation angle is 60 °. The scanning interval is 5 ° and a total

of 60 radial profiles are obtained for each scanning circle, lasting 135 s. Thanks to VAD scanning technology, aerosol signals and other hydrometeor signals can be identified (Wei et al., 2019). Detailed information about the system is presented in previous works (Yuan et al., 2020; Wang et al., 2017).

The 10.6 cm S-band Doppler weather radar deployed in Hefei is the first one of China's New Generation Weather Radar System (CINRAD), typed as CINRAD/SA developed based on the WSR-88D technology (He et al., 2012). It was installed on

the roof of a tower (31.867 °N, 117.258 °E) with a height of 116.5 m. During this thunderstorm, the CINRAD S-band data have a range gate spacing of 230 m and an azimuth spacing of nearly 1 °. Several preset scanning modes of operations are referred to as volume cover patterns (VCPs). In VCP-21 mode, each radar volume contains nine elevation scans: 0.5 °, 1.5 °, 2.4 °, 3.4 °, 4.3 °, 6.0 °, 9.9 °, 14.6 °, and 19.5 ° within 6 minutes.

**Table 1: Key parameters of CDWL, and DWR**

| Parameter | CDWL | DWR |
|---|---|---|
| Wavelength | 1.55 μm | 10.6 cm |
| Transmitter type | Pulsed (600 ns) | Pulsed (1.54 μs) |
| Transmitter power | 3 W (mean) | 650 kW (peak) |
| Pulse repetition rate | 10 kHz | 318 ~ 1300 Hz |
| Time resolution | 1 s | 0.1 s |
| Spatial resolution | 30 m | 1 km |
| Maximum detection range | 15 km | 230 km |
| Beam full divergence | 46 μrad | 0.99 ° |
| Azimuth scanning range | 0 ~ 360 ° | 0 ~ 360 ° |
| Zenith scanning range | 0 ~ 90 ° | 0 ~ 90 ° |






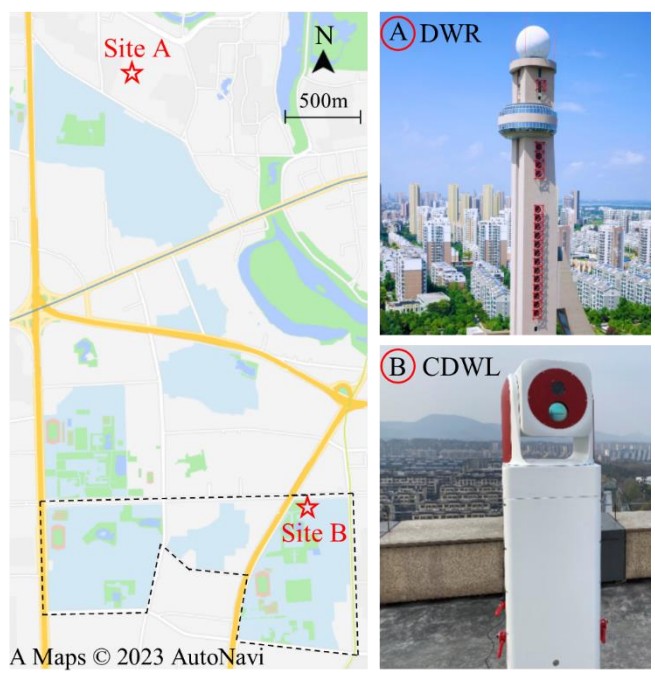

**Figure 2: The location of two instrument in map. Inside the dotted line is the campus of the USTC.**

In addition, the location of lightning data is obtained by a multi-station $E$-field sensor array consisting of 13 stations (Liu et al., 2021a; Liu et al., 2021b; Liu et al., 2018). Each station was equipped with a vertical $E$-field antenna (with 3 dB-bandwidth of 800 Hz to 300 kHz) and recorded the d$E$/dt (time derivative of the vertical $E$-field) for each lightning event. The atmospheric electricity sign convention is used for the $E$-field sensor. All recordings are synchronized with a GPS clock with sampling rate of 5 MHz. The location error of the networks is within 1 km and the error of estimated peak current is about 10% (Fan et al., 2020).

**2.2 Spaceborne instruments**

Fengyun-4 (FY-4) is the new generation of Chinese geostationary meteorological satellites with greatly enhanced capabilities for high-impact weather event monitoring, warning, and forecasting. FY-4 carries four new instruments: the Advanced Geosynchronous Radiation Imager (AGRI), the Geosynchronous Interferometric Infrared Sounder (GIIRS), the Lightning Mapping Imager (LMI), and the Space Environment Package (SEP) (Yang et al., 2017). The FY-4 measurements is applied to obtain cloud phase observation and to determine the occurrence of lightning activity in this study.

**2.3 ERA5 reanalysis data**

ERA5 is the fifth generation of the European Centre for Medium-Range Weather Forecasts (ECMWF) atmospheric reanalysis of the global climate. The ERA5 reanalysis assimilates a variety of observations and models in 4 dimensions (Hersbach et al., 2020). Since temperature changes can affect the phase of the hydrometeor in the atmosphere, the hourly temperature data from





the subdaily high-resolution-realization deterministic forecasts of ERA5 are used to infer atmospheric composition and phase
in this study.

## 3 Principle of CDWL detection

The wideband carrier-to-noise ratio (CNR) is the ratio of signal power to noise power. The accuracy of velocity estimation is
mainly determined by the CNR (Wang et al., 2017). The spectrum width is estimated by the ratio of total signal power to the
peak power value, and it represents velocity dispersion in a range bin. It can be broadened by windshear, turbulence, and
precipitation. Besides the CNR and spectrum width, normalized skewness is introduced to reveal how adverse weather
conditions affect the power spectrum in this work (Yuan et al., 2020; Yuan et al., 2021).

In order to improve the inversion probability of the wind vector in the weak signal regime, we apply a robust sine wave fitting
(RSWF) method which weights the contribution with a combination of CNR and fitting residual (Wei et al., 2020; Banakh et
al., 2010). In addition, since this study is more concerned with changes in the vertical direction within the thundercloud, power
spectrum of CDWL is an equivalent vertical detection spectrum derived from the radial spectra by compensating the Doppler
effect of the horizontal wind (Wei et al., 2021; Wei et al., 2019):

$$\tilde{V}_\perp = V_{LOS} - V_\parallel \cos(\varphi_0 - \theta_0) \sin\theta \qquad (1)$$

where $V_{LOS}$ is the line of sight (LOS) velocity, $V_\parallel$ is the horizontal wind speed, $\varphi_0$ is the elevation angle, $\theta_0$ is the horizontal
wind direction, $\theta$ is the azimuth angle of the lidar.

The TKEDR is a method for turbulence measurements using ground-based wind lidars (Sathe and Mann, 2013). TKEDR can
be estimated by fitting the azimuth structure function of radial velocity to a model prediction. In this work, this method is
applied to estimate the TKEDR in the VAD scanning mode. The method including error analysis are demonstrated in detail
(Banakh et al., 2017; Banakh and Smalikho, 2018). Note that the accuracy of wind and TKEDR mainly depends on CNR
(Wang et al., 2022b; Wang et al., 2021).

## 4 Experiments and analysis

### 4.1 Thunderstorm observation

On 30 April 2021, the Hefei region was affected by lightning activity in a thunderstorm. The images of lightning and hail
recorded from Hefei in Figure 2. Cloud-to-air channel had a greater luminosity than the upper channel. It can be clearly seen
that this is IC lightning. A few hail fell to the ground with diameter exceeded 10 mm.



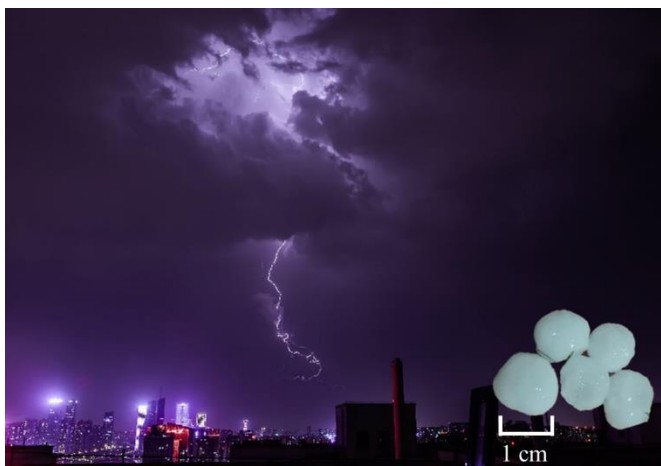


**Figure 2: IC lightning photograph captured in Hefei (31.817 ̊N, 117.222 ̊E) at 21:56 (LT) on 30 April 2021 by Dongting Zhao. The hail particles on the ground in the lower right.**

### 4.1.1 CDWL

The CDWL monitored the development of thundercloud over USTC from 20:00 to 22:00 local time (LT) (Figure 3). Figure

3a-g show the wideband CNR, spectrum width, spectrum skewness, horizontal wind speed, horizontal wind direction, vertical wind speed, and TKEDR measured by the CDWL. The 0 °C and -10 °C isotherms are from ERA5 data (Figure 3a-c). Due to the influence of updraft, large graupel and water drops are produced in thunderstorms, and the cloud develops rapidly, producing lightning and precipitation activities (Carey and Rutledge, 2000). The thundercloud over USTC occurs from 20:30 to 21:40 LT (Figure 3). As the laser cannot penetrate the region of the main updraft made up of large particles, the thundercloud

detected by CDWL turns into a 'V-cloud' (Figure 3a). Spectrum width broadening and skewness increase below the 0 °C isotherm and only spectrum width broadening below the -10 °C isotherm are observed during the period (Figure 3b and 3c). It means that there is a mixed region composed of multiple particles in the thundercloud below the 0 °C isotherm, that is, there is a melting layer. When the thundercloud appears, updrafts are detected near the surface (Figure 3d-f).

Figure 4 shows the results of meteorological data on the ground. The local atmospheric $E$-field is measured by a MEMS-

atmospheric ground $E$-field sensor (Figure 4a). It can record the $E$-field intensity induced by the thunderstorm. The local atmospheric $E$-field disturbances occur from 20:39 to 21:04 LT. During this period, lightning activities occur within a diameter of 10 km away from USTC measured by a multi-station $E$-field sensor array consisting of 13 stations. The short-term precipitation caused by the thunderstorm occurs between 20:54 and 21:06 LT (Figure 4b). And it was accompanied by the hail coming down.

During the descent of the thunderstorm cloudbase, the updrafts occurs near the surface (Figure 3d-f). And the local atmospheric $E$-field also begin to disturb, reaching a maximum before precipitation (Figures 4a and 4b). At this time, the horizontal wind direction of the thundercloud also changes, the lateral advection bursts at the cloudbase (Figure 3e). Then precipitation occurs,


it is should that the rain drops fall with charges (Marshall and Winn, 1982), the local atmospheric $E$-field gradually recovers to 0 (Figure 4a), the TKEDR increases exponentially near the surface (Figure 3g).

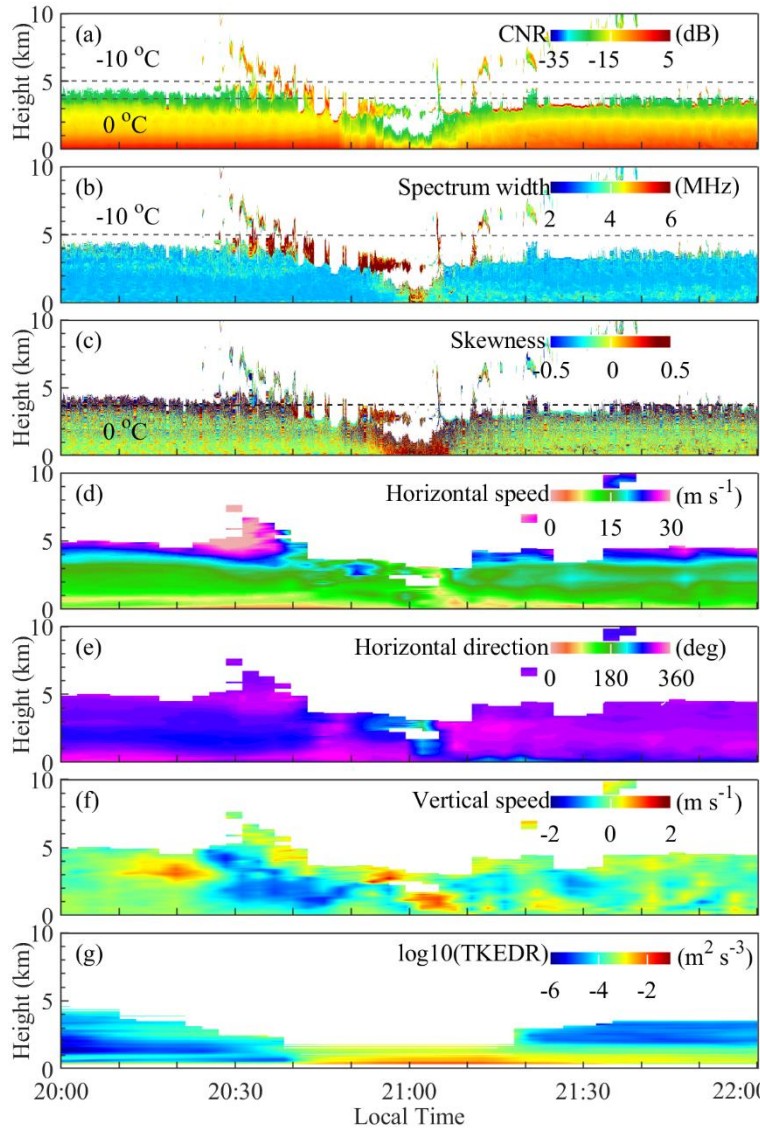


**Figure 3: Continuous observation results of CDWL, during the lightning activity in a thunderstorm event on 30 April 2021. a) The wideband CNR, the 0 °C and -10 °C isotherms are from ERA5, b) spectrum width, c) spectrum skewness, d) horizontal wind speed, e) horizontal wind direction, f) vertical wind speed, and g) TKEDR.**




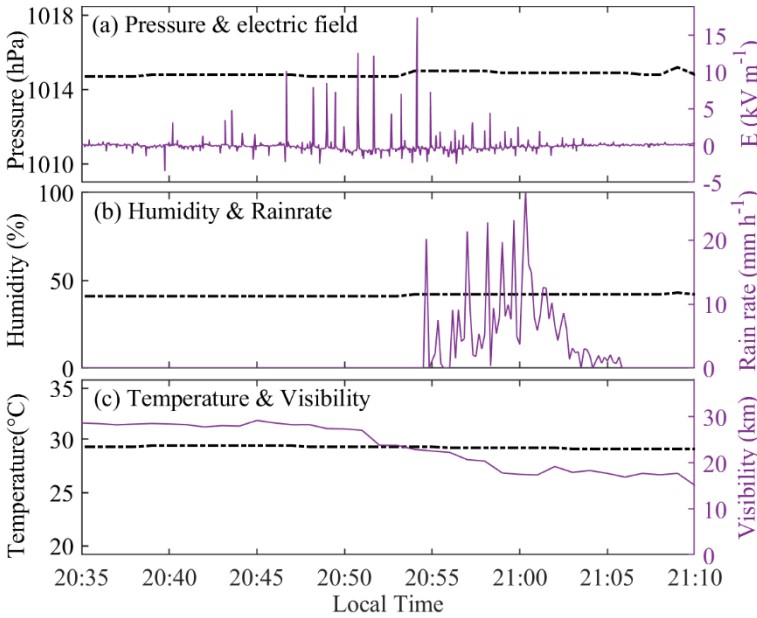

**Figure 4: Continuous observation results of pressure, local atmospheric E-field, humidity, rain rate, temperature and visibility on the ground level at during the lightning activity in a thunderstorm event on 30 April 2021. Temperature, visibility, humidity, rain rate and pressure provided by a Davis weather station. The atmospheric local *E*-field by MEMS -atmospheric ground *E*-field sensor.**

### 4.1.2 FY-4

The phase type of thunderstorm is provided by AGRI in FY4A satellites, with a spatial resolution of 4 km (Figure 5). Figure 4a is the cloud phase across the China region at 20:45 LT. The thunderstorm occurs in the target region, which is represented by the white square. The cloud phase distribution of thunderstorm regions at different times is shown in Figure 5b–g. The white pentagram indicates the location of the USTC. The orange arrows represent the wind vector.

During this time, the thunderstorm gradually moves southeast. Significant components of the ice phase, water phase, supercooled phase, and mixed phase can be seen in the thunderstorm. The ice phase occupies the center of the whole cloud, the water phase is in the outermost layer, and the supercooled phase and the mixed phase are the products of ice water mixing and exist at the interface of the ice phase and water phase. USTC is located at the boundary of the ice and water phase (Figure 5b-e). The thundercloud develops mainly vertically and expands less horizontally. The local atmosphere *E*-field disturbances also occur during this time. Subsequently, there are only ice phase particles in the thundercloud over USTC (Figure 5f and 5g). And the thundercloud mainly expands horizontally with the direction of the wind.





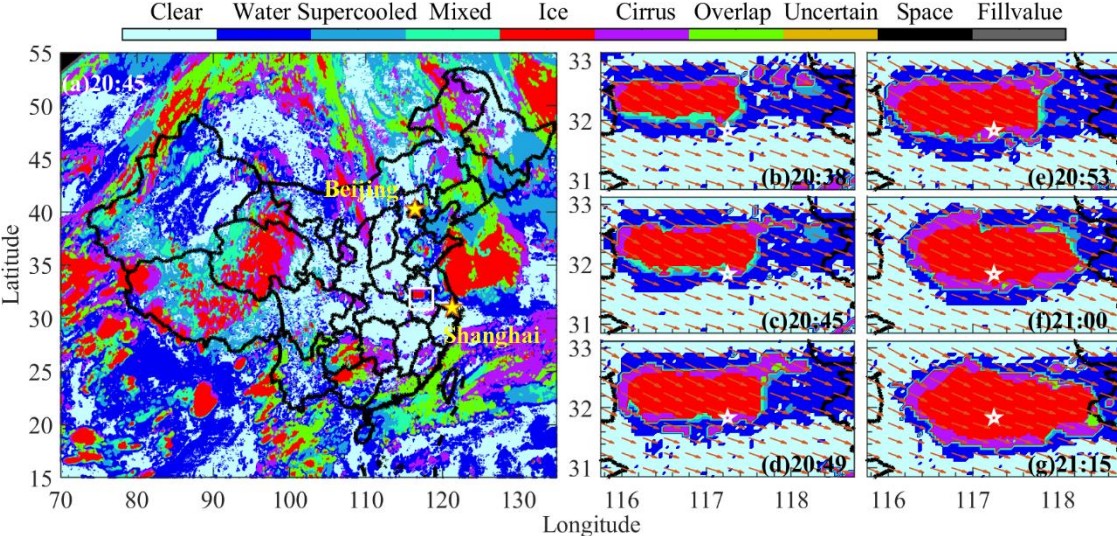

**Figure 5: The cloud phase type by FY-4 on 30 April 2021, local time. a) shows the cloud phase across the China region at 20:45 LT, the white square indicates the thunderstorm region. b)–g) show cloud phase distribution of thunderstorm at different times. The white pentagram and orange arrows represent the location of the USTC and the wind vector, respectively.**

### 4.1.3 DWR

Measurement results of the DWR give a more detailed and broader view of the thunderstorm development process. In addition, due to the VAD scanning mode by CDWL, the real cloud environment is different when higher clouds are detected, so measurement results of the DWR can also give a cloud environment changes over the USTC.

Figure 6 shows the reflectivity of DWR at each 6-min time step as the thundercloud is forming and developing. The location of USTC is marked with a pentagram. Lightning activity is more easily to occur in localized cells with higher reflectivity (Lang et al., 2004; Chmielewski et al., 2018). Consistent with observations from FY-4, the thunderstorm gradually moves to the southeast. At 20:39 LT, the cloud reaches USTC (Figure 6b), subsequently, the thundercloud continues to develop, and the reflectivity increases to a maximum of more than 55 dBZ (Figure 6c-e), most lightning appears during this time. After that, the reflectivity of the thundercloud gradually decreases (Figure 6e and 6f). Combined with precipitation data by Parsivel-2, it is indicated that hail is growing and falling at this time. When hail grows, the reflectivity is greater than 50 dBZ and no longer increases rapidly, but the strong reflectivity region increases (Williams, 2001). And as hail falls on the ground, the cloud dissipates and the reflectivity decreases rapidly. During the time of this thundercloud over USTC, the cloud reflectivity change less at the same time.



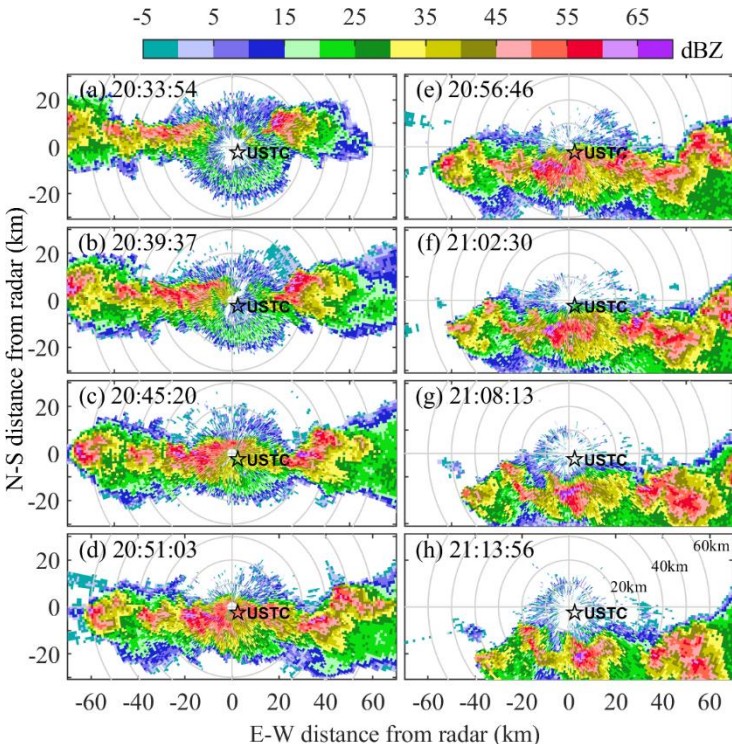

**Figure 6: Level-II reflectivity from Hefei S-band DWR observed at 6 °elevation angle for the thunderstorm development process. A**
**black pentagram represents the location of the USTC. The horizontal and vertical distance of 0 km and the diameter range of 10**
**km, 20 km, 30 km, 40 km, 50 km, and 60 km from the radar are drawn with gray lines.**

Figure 7a and 7b are the distance between the lightning occurrence point obtained by the multi-station $E$-field sensor array and
the USTC when the local atmospheric $E$-field disturbances occur (20:39-20:56 LT). The red solid circle represents the positive
cloud lightning, the black circle represents the negative cloud lightning, and the size of the mark represents the magnitude of
the peak current. Below the height of 8km, there are only negative cloud lightning at 3 km and 6 km height, the others are all
positive cloud lightning. Figure 7c is a vertical slice of the thundercloud over USTC detected by DWR, where the black line
represents the cloud boundary detected by CDWL, and blue and red lines represent the updrafts and downdrafts boundaries
detected by CDWL, respectively.

It can be seen that CDWL can only detect the outside region with the lower reflectivity of the thunderstorm. Updrafts below
the cloud exists in areas of apparently enhanced reflectivity, and downdrafts in areas of decreased reflectivity. The main updraft
region within the thundercloud undetected by CDWL with a reflectivity greater than 40 dBZ.

Stolzenburg et al. (1998) shows that there are differences in charge structure between the updraft region and the outside updraft
region of the thundercloud. In general, within convective updraft, the basic charge structure has four charge regions, alternating
in polarity, and the lowest one is positive. The charge structure outside the convective updraft is more complex, there are
typically at least six charge regions, alternating in polarity, and the lowest one is again positive. Therefore, the outside region
of the thundercloud detected by CDWL will be more electrically active (Stolzenburg et al., 2002).



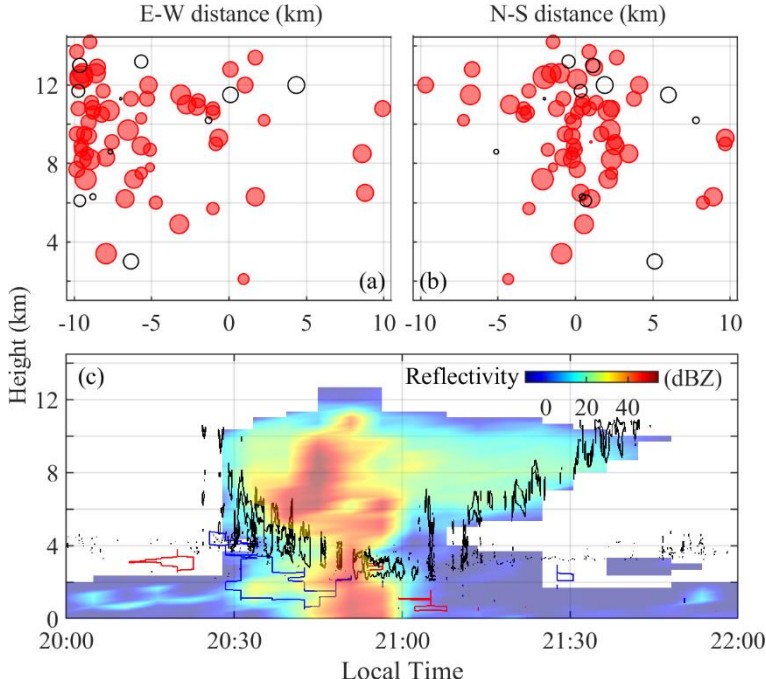

**Figure 7: During *E*-field disturbances, a) the east-west distance and b) the north-south distance between the lightning position detected by the multi-station E-field sensor array and the USTC, in which the red solid circle represents the positive cloud lightning,**
**the black circle represents the negative cloud lightning, and the size of the mark represents the magnitude of the peak current. c) The vertical slice of the thundercloud detected by DWR and the boundary of the thundercloud detected by CDWL.**

## 4.2 Further Discussion

Stolzenburg et al. (1998) shows that there are differences in charge structure between the updraft region and the outside updraft region of the thundercloud. The charge structure outside the convective updraft is more complex. Additional electric field

complexity and additional charge regions were found in the nonupdraft sounding (Stolzenburg et al., 2002). Therefore, a depth analysis of the power spectrum is performed to explore the structure of outside the convective updraft regions of thunderclouds detected by CDWL.

### 4.2.1 Distribution and motion characteristics

Under thunderstorm weather conditions, the received backscattering signal by CDWL could contain multiple components:

aerosol signal, water drop signal, ice crystal signal, graupel signal, raindrop signal and hail signal. From the Doppler power spectrum, two or more peaks can be observed if the velocities of mixed components are different. A multi-component Gaussian model is used to fit the multi-peak spectrum (Lottman and Frehlich, 1998; Wei et al., 2019):

$$S(f) = \sum I_n \exp\left[-\frac{(f-f_n)^2}{2\sigma_n^2}\right] \tag{2}$$

Where $f$, $I$, and $\sigma$ are Doppler frequency shift, peak intensity, and the spectrum width, the subscript $n$ represents components

such as aerosol, water drop, ice crystal, graupel, rain and hail, etc.



In order to investigate the composition, electrical properties, and motion characteristics of the melting layer in thundercloud, the CDWL power spectrum are separated into these particle spectrum following the procedure shown in Figure 8. The melting layer in the thundercloud is separated as the part of cold cloud. In addition, the CDWL typical spectrum width, skewness and vertical velocity of aerosol, raindrop, water drop, and graupel have been described in (Yuan et al., 2021; Yuan et al., 2020).

Firstly, the power spectrum is obtained from the CDWL raw data by using fast Fourier transform. The CNR, spectrum width, and skewness are derived from the power spectrum. Then, the cloud is extracted by CNR after range correction, with the HWCT method, where the CNR cloud threshold is -25 dB (Wang et al., 2021). In the next part, it is separated into warm and cold cloud spectrum by the internal cloud temperature. Finally, Doppler spectra of different components are determined by spectrum width and skewness. Note that the rain and hail are not directly separated in this work, and rain/hail are categorized

in the water classification are shown together in Figure 9. In the next work, we will further distinguish between rain and hail spectrum.

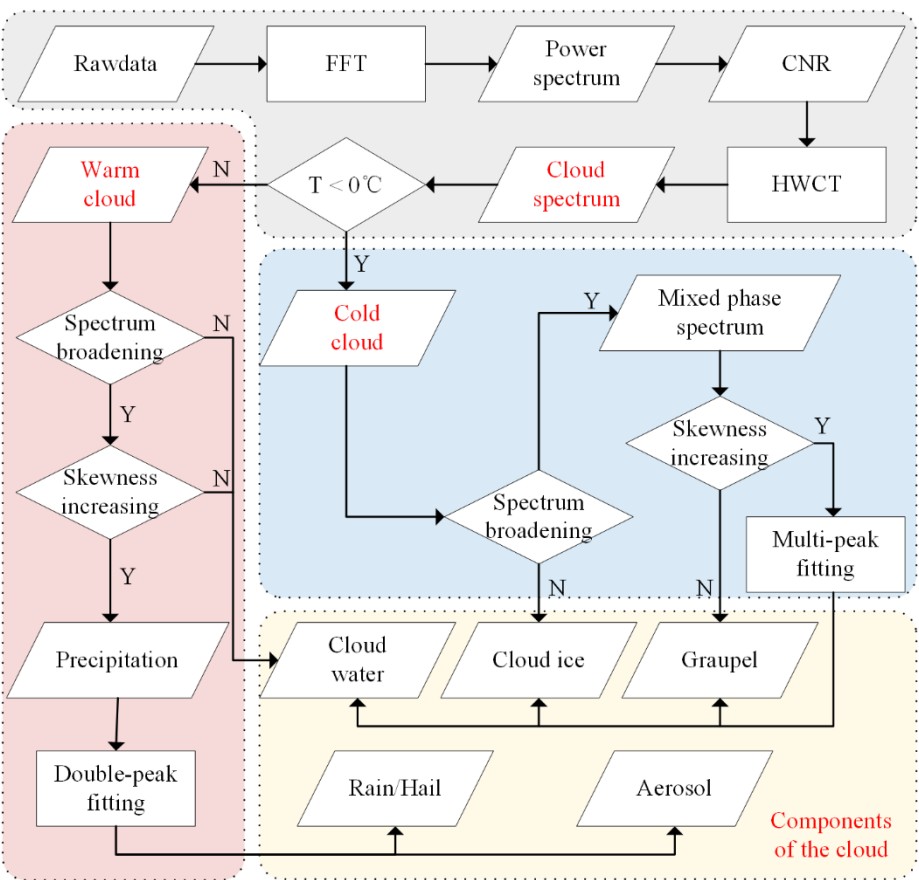

**Figure 8: The process of separating multi-component particles in the thundercloud.**





**Figure 9: Separation results of aerosol, ice, water and graupel. (a) Aerosol CNR; (b) Aerosol spectrum width; (c) Aerosol horizontal speed; (d) Aerosol horizontal direction; (e) Aerosol vertical speed. (f) Ice CNR; (g) Ice spectrum width; (h) Ice horizontal speed; (i) Ice horizontal direction; (j) Ice vertical speed. (k) Water CNR; (l) Water spectrum width; (m) Water horizontal speed; (n) Water horizontal direction; (o) Water vertical speed. (p) Graupel CNR; (q) Graupel spectrum width; (r) Graupel horizontal speed; (s) Graupel horizontal direction; (t) Graupel vertical speed.**

Figure 9 shows the wideband CNR, spectrum width, horizontal speed, horizontal direction, and vertical speed of different particles. Cloud ice spectrum width is slightly larger than cloud water and aerosol spectrum width due to ice crystal is more prone to multiple scattering (Figure 9b, 9g and 9l). Cloud ice mainly occurs above 5 km height, due to the action of horizontal pressure-gradient force, its horizontal velocity is much faster than cloud water at low heights (Figure 9h and 9m). After the initial disturbance of the *E*-field (at 20:40 LT), the ice around 5 km has a significant decline speed (Figure 9j).

Before precipitation, spectrum width of cloud water above 3 km is broadened (Figure 9l), which is inside the melting layer, the water drops are mixed with the melted graupel, resulting in the separated cloud water spectra is broader than the typical





cloud water spectra. At this time, cloud water also has a small rising speed (Figure 9o). During precipitation, the precipitation particles carry charges and form lower positive charge centers (Marshall and Winn, 1982; Reynolds et al., 1957). There is a slight disturbance in the atmospheric *E*-field (Figure 4a). The spectrum width of the precipitation particle is much greater than

5 MHz (Figure 9l), has a distinct downward speed (Figure 9o). After precipitation, the CNR of cloud water around 3 km becomes higher (Figure 9k) because the cloud layer is thicker, the laser cannot fully penetrate, and the received backscattering signal is superimposed.

Graupel mainly appeared in 20:40-21:05 LT, which is consistent with the disturbance period of the local atmospheric *E*-field, indicating that graupel is the main reason for *E*-field changes. During precipitation, the spectrum width of the graupel is much

greater than 5 MHz (Figure 9q), has a distinct downward speed within the thundercloud, and during precipitation, hail with the raindrops fall together (Figure 9o).

**4.2.2 The effects caused by the lightning**

In order to investigate the effect of lightning on the surrounding thundercloud environment, the CDWL power spectrum are selected when the lightning number is larger than 2 detected by the multi-station *E*-field sensor array (Figure 10). Positive

velocities represent downward motion and negative velocities represent upward motion. It is noted that this velocity has removed the influence of horizontal wind and can be approximately regarded as vertical velocity.

Figure 10a1 shows the power spectrum at 20:39:13 LT with a slight local atmospheric *E*-field disturbance. The thundercloud is in the initial electrification stage. The cloud below the -10 ℃ isotherm has a significantly broadened spectrum, and there is a graupel with downward speed, as shown in Figure 10a2, which is a typical graupel power spectrum. Figure 10a3 and a4 show

the typical power spectrum of ice crystal and liquid water in a cloud, respectively. That should be the ice crystals formed when failing graupel collided with rising supercooled liquid water (Wei et al., 2019). Figures 10a5 shows the double-peak power spectrum when liquid water is mixed with aerosol. The cloud below the 0 ℃ isotherm borders the aerosols, the spectrum has not broadened, and there is no melting layer.

Figures 10b1 and c1 show the power spectrum when a CG lightning. The highest height of the cloud detected by CDWL is

bigger than 0, and according to the movement trend here, there is likely to be a balance layer of vertical velocity = 0 m/s above the height of 5 km (Yuan et al., 2021). Spectrum width of the cloud is very broad, and there is obvious melting layer. As shown in Figure 10b2, the spectrum width > 12 MHz, in addition to the existing graupel and liquid water drops in this region, there is also a graupel vertical velocity > 5 m/s. This graupel is separated from the cold cloud particle (Figure 8) and is melting below the 0 ℃ isotherm. Then, this graupel mixed with liquid water, and the velocity shifts downward as the whole cloud

moves down (Figure 10b3-b5). Figure 9c1 shows the power spectrum before surface precipitation. It can be clearly seen that there also has graupel with a speed > 5 m/s in the melting layer (Figure 10c2-5). At the same time, there has graupel with a velocity < 5m/s, which is gradually melted during the falling process and the speed decreases.

Figure 10d1 is also the power spectrum of IC lightning before surface precipitation. The cloud below the 0 ℃ isotherm has a significant broadening spectrum, but due to the imminent surface precipitation, there has graupel with large velocity on the





right side of the power spectrum, which is gradually melted into droplet during the falling process, the speed decreases, and then falls to the surface to form precipitation (Figure 10d2-5).

Figures 10e1 and f1 show the power spectrum during precipitation. A significant difference in velocity between the upper and lower of the detected cloud can be seen due to the broadening of the spectrum due to raindrops during precipitation. As shown in Figure 10e3, the typical intra-cloud precipitation spectrum with hail, which is formed by liquid cloud droplets, liquid

raindrops, and graupel.



**Figure 10:** (a1) - (f1) are the Doppler power spectrum intensity at different times of by lightning in the thundercloud detected by CDWL. The grey dotted line indicates that the vertical speed is 0 m s-1. The orange and red lines represent the -10 ℃ and 0 ℃ isotherms, respectively. Positive velocities represent downward motion and negative velocities upward motion. (a2) - (f5) are the


**specific spectra and multi-component Gaussian fitting curves at different heights. It is normalized with to the peak of the power spectrum as the maximum, the H, Sw and Sk are the Height, spectrum width and skewness of the raw spectrum. The blue dot markers stand for the raw data of spectra. Black lines represent fitting results. The purple shadow and pink shadow represent separated graupel and ice crystal components, respectively. The blue shadow and yellow shadow represent separated liquid water and aerosol components, respectively.**

It can be seen that CG lightning has a broader spectrum than IC lightning, and there has graupel with a velocity > 5 m/s during CG lightning. It should be that the height of IC lightning occurs is higher, and has less impact on the cloud environment below 5 km height. However, the CG lightning has an impact on the whole thundercloud environment.

Yuan et al. (2020) has been shown that lightning causes Doppler spectrum broadening, both in the balance layer and in melting layer. It can be seen in Figure 10, CG lightning has a broader spectrum than IC lightning, and there has graupel with a velocity > 345 5 m/s during CG lightning. Thus, the additional graupel particles with a velocity> 5 m/s should be caused by lightning, rather than the particle in the melting layer formed after the IC channel is heated. Moreover, most of the IC lightning occurs height is higher, and has less impact on the cloud environment below 5 km height. However, the CG lightning has an impact on the whole thundercloud environment.

In the absence of lightning occurrence, the melting layer is detected by CDWL. Furthermore, to identify and separate different 350 particles in the thundercloud, which provides a new ways to study the electrical process of the thundercloud and the impact of the surrounding environment. Meanwhile, the detection of graupel in the melting layer and precipitation also provide more detailed information on the development process of the thunderstorm.

**5 Conclusions**

In this paper, observations of the thunderstorm were reported based on the CDWL, DWR, FY-4, and other ground instruments. 355 The formation, rapid growth, and dissipation of thunderclouds were monitored and analysed. Although the cloud was not penetrated completely by CDWL, the broadened spectrum width and increased skewness below the 0 ℃ isotherm in outside the convective updraft region of the thunderstorm was observed. The changing characteristics of the particle velocity, phase, and component in this region were also detected through the power spectrum analysis. Combined with the lightning detected by multiple sensors, it was found that when there has additional graupel with a speed greater than 5 m/s in the thundercloud 360 when a CG lightning within 10 km nearby. It was proved that CDWL has the ability to observe the composition and motion characteristics of thunderstorms. At present, we cannot detect the main updraft region of the thundercloud. In future work, we will try to improve cloud penetration ability by increasing the power of the laser to study the composition and motion characteristics of particles in the upper thundercloud, as well as the velocity changes in the cloud under the IC lightning occurs, and achieve early warning of thunderstorm activity and hail precipitation. We also plan to integrate polarization detection into 365 the lidar system (Wang et al., 2021), and conduct more detailed observations of particle phase changes and charge structure in thunderclouds.



*Data availability.* The ERA5 data sets are publicly available from the ECMWF website at https://cds.climate.copernicus.eu/cdsapp#!/home (last access: 10 April 2023). The Fengyun-4 satellites data are available on the National Satellite Meteorological Center (NSMC) website at http://satellite.nsmc.org.cn/portalsite/Data/Satellite.aspx (l

last access: 10 April 2023). The CDWL and DWR data can be downloaded from https://figshare.com/articles/dataset/A_thundercloud_lidar_results_during_the_experiment/20326350 (last access: 10 April 2023), https://figshare.com/articles/media/raw_data_video_of_CDWL_during_this_experiment/21590433 (last access: 10 April 2023) and https://figshare.com/articles/dataset/A_thundercloud_rader_results_during_the_experiment/20326377 (last access: 10 April 2023). The lightning location detected by a Micro-Electro-Mechanical System (MEMS)-atmospheric ground

E-field sensor data can be downloaded from https://doi.org/10.6084/m9.figshare.20588385 (last access: 10 April 2023). The precipitation data by second-generation particle size and velocity, Parsivel-2 can be downloaded from https://doi.org/10.6084/m9.figshare.22256617 (last access: 10 April 2023).

*Author contributions.* H.X., K.W., and T.W. planned the campaign; K.W., T.W., X.H. F.L. Y.Z and W.D. performed the measurements; K.W., T.W., J.Y. and X.H. analyzed the data; K.W. wrote the manuscript draft; G.L., B.Z., and H.X. reviewed

and edited the manuscript. All authors have read and agreed to the published version of the manuscript.

*Competing interests.* The authors declare that they have no conflict of interest.

*Copyright statements.* The copyright statement will be included by Copernicus.

*Acknowledgements.* We thank photographer Zhao Dongting, who takes pictures of IC lightning and hail particles falling to the ground; The National Satellite Meteorological Center of the China Meteorological Administration for providing support with

the observational data; The European Centre for Medium-Range Weather Forecasts for providing support with atmospheric reanalysis data.

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
