# Peer review of "Thundercloud structures detected and analyzed based on coherent Doppler wind lidar"

_Atmospheric Measurement Techniques, 2023_

## Author Comment (AC1)

We would like to thank the reviewers for their valuable comments and suggestions. We have considered all comments carefully which helped us significantly to improve our manuscript. Following the reviewers' comments and suggestions, we revised the manuscript. Our responses to the reviewers' comments are listed below in blue fonts and the changes in manuscript are listed in *blue italic fonts*.

**Anonymous Referee #1**

Wu et al. presented an interesting study showing observations and characterization of the thunderstorm based on measurements from coherent Doppler wind lidar (CDWL), Doppler weather radar and other instruments. Application of CDWL to study the atmosphere during thunderstorm is a relatively novel topic in remote sensing. I recommend acceptance for publication in AMT after minor revision.

Thanks for your careful and thoughtful comments. We revised the manuscript according to your suggestions.

1. VAD scanning mode is used for the CDWL in the experiment. The observation path is titled at an elevation angle of 60 degrees rather than vertical, why?

Thanks for this comment. Vertical detection can only capture cloud signals vertically. In the presence of thick clouds, such as the thundercloud in this work, the laser is unable to penetrate through the cloud layer. However, with the utilization of VAD scanning, not only can we acquire three-dimensional wind field information within the cloud, but also, due to the laser entering the cloud at a 60-degree elevation angle, it can effectively penetrate more cloud structures and detect a higher of cloud signals.

2. How extensive is attenuation of lidar signals?

Thank you for your concerns. The black line in Figure 8c represents the cloud boundary detected by CDWL. Compared with weather radar, lidar can only detect the outside updraft region of the thundercloud due to insufficient laser power. Within thick convective clouds, only cloud signals within the upper 2 km of the cloudbase can be detected by CDWL.

3. CDWL cannot penetrate thick clouds so the results here can only reveal (with high uncertainty) the lower part of the thundercloud (mostly <5 km). Please summarize the limitations of the CDWL measurements.

Thank you for your concerns. Compared with other radars, the power of laser by CDWL is not large enough to penetrate thicker clouds. And CDWL does not have polarization detection at present, it cannot make more detailed observations of the particle phase, but we are integrating the polarization detection into the lidar system. In addition, it is a ground-based remote sensing instrument that cannot observe the entire formation and development of thunderstorms from a horizontal perspective.

4. During thunderstorms, the strength of vertical wind shear is correlated with the strength of thunderstorms. Some regrets fail to discuss the impact from turbulence on the velocity spectrum.

Thank you very much for your kind reminder. Turbulent activity within the cloud can provide relative velocity differences for aquatic particles of different masses, affecting the collision speed of graupel and ice crystals in the mixed-phase region, and substantially participating in the thunderstorm initiation process. This study mainly inferred the composition and velocity change based on the Doppler velocity spectrum of CDWL, that is, the velocity change caused by turbulence has been included. In

addition, CDWL only detects the outside updraft region of the thunderstorm. Therefore, the effect of turbulence on the velocity spectrum is not discussed in detail.

**Changes:**

In line 167-168, "*Currently, CDWL is capable of accurately assessing the turbulence changes from the surface to the cloudbase of the thunderstorm.*"

There are still some grammatical problems need to be carefully checked.

1. Line 162-163: "The images of lightning and hail recorded from Hefei in Figure 2" is not a readable sentence.

Thank you for your kind reminder. We have revised corresponding sentences according to your suggestions.

**Changes:**

In line 192-193, "*The images of lightning and hail recorded in Hefei are shown in Figure 3.*"

2. Line 195-196: "at" needs to be deleted.

Thank you for your kind reminder. We have revised corresponding sentences according to your suggestions.

**Changes:**

In line 225-226, "*Figure 4: Continuous observation results of pressure, local atmospheric E-field, humidity, rain rate, temperature and visibility on the ground level during the lightning activity in a thunderstorm event on 30 April 2021.*"

3. Line 199: replace "in" with "on board".

Thank you for your kind reminder. We have revised corresponding word according to your suggestions.

**Changes:**

In line 229, "*The phase type of thunderstorm is provided by AGRI on board FY4A satellites, with a spatial resolution of 4 km (Figure 6).*"

4. Line 203: replace "of" with "including".

Thank you for your kind reminder. We have revised corresponding word according to your suggestions.

**Changes:**

In line 233-234, "*Significant components including the ice phase, water phase, supercooled phase, and mixed phase can be seen in the thunderstorm.*"

5. Line 215-216: the sentence "the real cloud environment is different when higher clouds are detected, so measurement results of the DWR can also give a cloud environment changes over the USTC" can't be understood.

Thank you for your kind reminder. We have revised corresponding sentences according to your suggestions.

**Changes:**

In line 245-247, "*In addition, due to the utilization of VAD scanning mode by CDWL to investigate cloud environments from various azimuth angles, the measurement results of the DWR can also indicate*

*changes in the horizontal cloud environment over the USTC.*"

6. Line 240: "exists" should be "exist"

Thank you for your kind reminder. We have revised corresponding word according to your suggestions.

**Changes:**

In line 269-270, "*Updrafts below the cloud exist in areas of apparently enhanced reflectivity, and downdrafts in areas of decreased reflectivity.*"

7. Line 333: "0 m s-1".

Thank you for your kind reminder. We have revised corresponding sentences according to your suggestions.

**Changes:**

In line 347, "*The grey dotted line indicates that the vertical speed is 0 m $s^{-1}$.*"

8. Line 345: "most of the IC lightning occurs height is higher" is not a readable sentence.

Thank you for your kind reminder. We have revised corresponding sentences according to your suggestions.

**Changes:**

In line 360-361, "*Moreover, most of the IC lightning occurs at higher height, and has less impact on the cloud environment below 5 km height.*"

9. Line 358: "Combined with the lightning detected by multiple sensors, it was found that when there has additional graupel with a speed greater than 5 m/s in the thundercloud when a CG lightning within 10 km nearby." is not a readable sentence.

Thank you for your kind reminder. We have revised corresponding sentences according to your suggestions.

**Changes:**

In line 373-374, "*Combined with the lightning detected by multiple sensors, it was found that CG lightning occurs within a 10 km radius when there is additional graupel with a speed greater than 5 m/s in the thundercloud.*"

**Anonymous Referee #2**

The paper presents a novel method to study thundercloud structures using coherent Doppler lidar wind. A particular study case at Hefei (China) is used to illustrate the potential of the method. The method is novel and valuable for publication. In general, I agree with all the comments made by the previous referee. But I add a major revision that need to be addressed and is related with the structure of the paper. Basically, it is difficult to find what is the new methodology. A flow chart (Figure 8) is presented with details of the methodology are given in section 4.2.1, and they should be moved to a methodology section before. I would recommend joining with the section 'Principle of CDWL detetion' and make a more detailed and consistent methodology section, highlighting the novelties versus other developments/applications.

Thanks for your professional comments and pointing out the shortcomings in the manuscript. We

revised the structure and methodology of the manuscript according to your suggestions.

Changes:

In line 146-189, "**3 Principle of CDWL detection**

**3.1 Characteristics of the power spectrum**

*The wideband carrier-to-noise ratio (CNR) is the ratio of signal power to noise power. The accuracy of velocity estimation is mainly determined by the CNR (Wang et al., 2017). The spectrum width is estimated by the ratio of total signal power to the peak power value, and it represents velocity dispersion in a range bin. It can be broadened by windshear, turbulence, and precipitation. Besides the CNR and spectrum width, normalized skewness is introduced to reveal how adverse weather conditions affect the power spectrum in this work (Yuan et al., 2020; Yuan et al., 2021).*

*In order to improve the inversion probability of the wind vector in the weak signal regime, we apply a robust sine wave fitting (RSWF) method which weights the contribution with a combination of CNR and fitting residual (Wei et al., 2020; Banakh et al., 2010). In addition, since this study is more concerned with changes in the vertical direction within the thundercloud, power spectrum of CDWL is an equivalent vertical detection spectrum derived from the radial spectra by compensating the Doppler effect of the horizontal wind (Wei et al., 2021; Wei et al., 2019):*

$$\tilde{V}_\perp = V_{LOS} - V_\| \cos(\varphi_0 - \theta_0) \sin\theta \qquad (1)$$

*where $V_{LOS}$ is the line of sight (LOS) velocity, $V_\|$ is the horizontal wind speed, $\varphi_0$ is the elevation angle, $\theta_0$ is the horizontal wind direction, $\theta$ is the azimuth angle of the lidar.*

**3.2 Calculation of turbulence**

*The TKEDR is a method for turbulence measurements using ground-based wind lidars (Sathe and Mann, 2013). TKEDR can be estimated by fitting the azimuth structure function of radial velocity to a model prediction. In this work, this method is applied to estimate the TKEDR in the VAD scanning mode. The method including error analysis are demonstrated in detail (Banakh et al., 2017; Banakh and Smalikho, 2018). Note that the accuracy of wind and TKEDR mainly depends on CNR (Wang et al., 2022b; Wang et al., 2021). The thundercloud atmospheric motion is a complex pattern of combined updrafts and downdrafts, exhibiting continuous turbulence at different scales (Bryan et al., 2003; Feist et al., 2019). Currently, CDWL is capable of accurately assessing the turbulence changes from the surface to the cloudbase of the thunderstorm.*

**3.3 Thundercloud composition identification algorithm**

*Under thunderstorm weather conditions, the received backscattering signal by CDWL could contain multiple components: aerosol signal, water drop signal, ice crystal signal, graupel signal, raindrop signal and hail signal. From the Doppler power spectrum, two or more peaks can be observed if the velocities of mixed components are different. A multi-component Gaussian model is used to fit the multi-peak spectrum (Lottman and Frehlich, 1998; Wei et al., 2019):*

$$S(f) = \sum I_n \exp\left[-\frac{(f-f_n)^2}{2\sigma_n^2}\right] \qquad (2)$$

*Where $f$, $I$, and $\sigma$ are Doppler frequency shift, peak intensity, and the spectrum width, the subscript $n$ represents components such as aerosol, water drop, ice crystal, graupel, rain and hail, etc.*

*In order to investigate the composition, electrical properties, and motion characteristics of the melting layer in thundercloud, the CDWL power spectrum are separated into these particle spectra following the procedure shown in Figure 1. The melting layer in the thundercloud is separated as the part of cold cloud. In addition, the CDWL typical spectrum width, skewness and vertical velocity of aerosol, raindrop, water drop, and graupel have been described in (Yuan et al., 2021; Yuan et al., 2020).*

*Firstly, the power spectrum is obtained from the CDWL raw data by using fast Fourier transform. The CNR, spectrum width, and skewness (Sec.3.1) are derived from the power spectrum. Then, the cloud is extracted by CNR after range correction, with the HWCT method, where the CNR cloud threshold is -25 dB (Wang et al., 2021). In the next part, it is separated into warm and cold cloud spectrum by the internal cloud temperature. Finally, Doppler spectra of different components are determined by spectrum width and skewness. Note that the rain and hail are not directly separated in this work, and rain/hail are categorized in the water classification are shown together in Figure 9. In the next work, we will further distinguish between rain and hail spectrum.*

[Figure]

*Figure 2: The process of separating multi-component particles in the thundercloud.*"

Apart of that, I would like to add a few minor comments:

- Between lines 39-60, there is a long discussion with lack of references. This is in the introduction section, and I believe that the discussion is based on previous studies that must be cited.

*Thank you for your kind reminder. We have revised corresponding sentences according to your suggestions.*

**Changes:**

*In line 37-53, "The NIC mechanism is thought to be primarily responsible for the thunderstorm discharges. The theory is based on experiments conducted in Japan by Takahashi (1978) and in the UK by Jayaratne et al. (1983). Observations revealed that the magnitude and polarity of the charging process depend on the water content of the cloud and the ambient temperature. During collisions graupel charge positively at higher temperatures, at both low and high water content, and they charge negatively at low temperatures and at intermediate water content (Takahashi, 1978). Consider a typical cloud with a liquid*

*water content of approximately 1 g/m³, the positive charge center is located above the negative one, and the negative one is very shallow, approximately 1 km in thickness, and located in a region of -15 to -10 ℃ isotherm. Below the negative charge center is a small positive charge pocket (Cooray, 2015). As the graupel particles fall from greater heights through the clouds, they collide with ice crystals that are being carried upward in updrafts (Low and List, 1982; Hallett et al., 1978; Beard, 1976). If the temperature is below approximately -15 to -10 ℃, the graupel particles charge negatively and the ice crystals positively. The light positively charged ice crystals travel upward along the updraft, leaving the positive charge at a higher location in comparison with the negatively charged falling graupel particles (Williams, 1989, 1988). As the graupel particles fall further, the temperature increases and the graupel particles start to charge positively. Thus, there is a region below the height of the isotherms -15 to -10 ℃ where graupel particles are positively charged. This is the basis of the positive charge pocket located below the negative charge center. This creates the observed tripolar structure of the cloud (Bruning et al., 2014; Williams, 1989, 2001; Bruning et al., 2010). It also explains why the main negative charge center is located in the region of the -15 and -10 ℃ isotherm (Cooray, 2015).*"

- I miss details of the CDWL. Is the instrument commercial or home-made? In the last case, more details about its configuration are needed.

Thank you for your concerns. The CDWL is homemade. We have added more details, as your suggestion.

**Changes:**

In Table 1: Key parameters of CDWL, and DWR

| Parameter | CDWL | DWR |
|---|---|---|
| Wavelength | 1.55 μm | 10.6 cm |
| Transmitter type | Pulsed (600 ns) | Pulsed (1.54 μs) |
| Transmitter power | 3 W (mean) | 650 kW (peak) |
| Pulse repetition rate | 10 kHz | 318 ~ 1300 Hz |
| Diameter of telescope | 80 mm | - |
| AOM frequency shift | 80 MHz | - |
| Time resolution | 1 s | 0.1 s |
| Spatial resolution | 30 m | 1 km |
| Maximum detection range | 15 km | 230 km |
| Antenna diameter | 10 cm | 8.54 m |
| Beam full divergence | 46 μrad | 0.99 ° |
| Azimuth scanning range | 0 ~ 360 ° | 0 ~ 360 ° |
| Zenith scanning range | 0 ~ 90 ° | 0 ~ 90 ° |

In line 100-101, "*More detailed parameters and applications of the CDWL are introduced in previous work (Wei et al., 2020; Wang et al., 2019a).*"

- Conclusion section poorly reflects the novelty of the work, what are the limitations found and what is the future work.

Thanks for your positive comments. We have rewritten some conclusions according to your suggestions.

**Changes:**

In line 368-381, "*In this paper, a novel method for identifying thundercloud particles based on*

*CDWL power spectrum analysis was proposed. And observations of the thunderstorm were reported based on the CDWL, DWR, FY-4, and other ground instruments. The formation, rapid growth, and dissipation of thunderclouds were monitored and analysed. While CDWL did not completely penetrate the cloud, it successfully detected a broadened spectrum width and increased skewness below the 0 ℃ isotherm, outside the convective updraft region of the thunderstorm. Moreover, this region exhibited significant variations in particle velocity, phase, and composition. Combined with the lightning detected by multiple sensors, it was found that CG lightning occurs within a 10 km radius when there is additional graupel with a speed greater than 5 m/s in the thundercloud. These findings validate the capacity of CDWL to observe the composition and motion characteristics of thunderstorms. At present, we cannot detect the main updraft region of the thundercloud. In future work, we will try to improve cloud penetration ability by increasing the power of the laser. This will enable us to study the composition and motion characteristics of particles in the upper thundercloud, as well as the velocity changes in the cloud when IC lightning occurs. Additionally, we aim to conduct comprehensive observations of the electrification process and lifecycle of thunderstorms. We also plan to integrate polarization detection into the lidar system (Qiu et al., 2017), and perform more detailed observations of particle phase changes and charge structure in thunderclouds."*

**References:**

Cooray, V.: Charge Generation in Thunderclouds and Different Forms of Lightning Flashes, in: An Introduction to Lightning, Springer Netherlands, Dordrecht, 79-89, https://doi.org/10.1007/978-94-017-8938-7_6, 2015.

Low, T. B. and List, R.: Collision, Coalescence and Breakup of Raindrops. Part I: Experimentally Established Coalescence Efficiencies and Fragment Size Distributions in Breakup, Journal of the Atmospheric Sciences, 39, 1591-1606, https://journals.ametsoc.org/view/journals/atsc/39/7/1520-0469_1982_039_1591_ccabor_2_0_co_2.xml, 1982.

Hallett, J., Sax, R. I., Lamb, D., and Murty, A. S. R.: Aircraft measurements of ice in Florida cumuli, Quarterly Journal of the Royal Meteorological Society, 104, 631-651, https://doi.org/10.1002/qj.49710444108, 1978.

Beard, K. V.: Terminal Velocity and Shape of Cloud and Precipitation Drops Aloft, Journal of the Atmospheric Sciences, 33, 851-864, https://journals.ametsoc.org/view/journals/atsc/33/5/1520-0469_1976_033_0851_tvasoc_2_0_co_2.xml, 1976.

Williams, E. R.: The Electrification of Thunderstorms, Scientific American, 259, 88-99, https://www.jstor.org/stable/24989265, 1988.

Williams, E. R.: The tripole structure of thunderstorms, Journal of Geophysical Research: Atmospheres, 94, 13151-13167, https://doi.org/10.1029/JD094iD11p13151, 1989.